# The Health Consequences of Social Mobility in Contemporary China

**DOI:** 10.3390/ijerph15122644

**Published:** 2018-11-26

**Authors:** Fei Yan, Guangye He, Yunsong Chen

**Affiliations:** 1Department of Sociology, Tsinghua University, Beijing 100084, China; feiyan@tsinghua.edu.cn; 2Department of Sociology, Nanjing University, Nanjing 210008, China

**Keywords:** self-rated health, social mobility, China, upward mobility, downward mobility

## Abstract

Although numerous studies have shown the importance of an individual’s socioeconomic status on his or her self-rated health status, less well-known is whether self-perceived class mobility, a measure highly correlated with an individual’s de facto social class and past mobility experiences, affects self-rated health. In this paper, we attempt to fill the gap by examining how perception of class mobility is associated with self-rated health. Using eight waves of Chinese General Social Survey data spanning the years 2005 to 2015, we conducted an analysis at the micro (individual) level and the macro (provincial) level. Analyses at both levels yielded consistent results. At the individual level, we employed ordered logistic regression and found that the perception of experiencing downward mobility was associated with significantly lower self-rated health in both rural and urban areas compared with those who consider themselves to be upwardly mobile or immobile. At the provincial level, the findings from static panel analysis further revealed that there is a positive relationship between the self-perceived class mobility and self-rated health level.

## 1. Introduction

Research on the relationship between an individual’s social class and health has been plentiful in the last few decades, and much scholarly discussion has been devoted to the relative explanatory power of two main hypotheses: social causation and health selection [1,2,3,4]. 

The social causation hypothesis argues that health is affected by socially determined structural factors. Family income, an individual’s employment status, childhood living conditions, and mobility trajectories are common variables that help measure how an individual’s social mobility affects his or her health. Early research indicated that upward social mobility is accompanied by better health, while downward social mobility leads to poorer health [5,6,7,8]. However, recent research seems to challenge these results, showing that while downward social mobility does indeed predict negative health, upward social mobility is not always accompanied by improved health [9,10]. One study even proposed that upward mobility would result in poorer health and downward mobility to better health [11], based on the observation that individuals and families who experience rapid upward mobility may at the same time become socially isolated and excluded from both their original class and their new class group [12,13], thus increasing the risk of negative physical and mental health. Sorokin also proposed that although societies with inter-class mobility are more dynamic and active, individuals in such societies usually experience more life stress and low social trust [14]. Moreover, individuals in socially mobile societies find it harder to form intimate relationship, and the weak social ties that result can lead to more mental illnesses among adults.

The health selection hypothesis maintains that social mobility is affected by health—the healthier the individual, the more likely his or her upward social advance; the less healthy the individual, the more likely his or her downward social movement [15]. According to this argument, health does play an important role in social mobility, affecting education, performance in the labor market, and intergenerational mobility [16,17,18,19]. Previous studies have shown that parents’ health status is positively correlated to their next generations’ income level; however, such results are applicable for low-income families and are not significant in high-income families [17]. 

Meanwhile, the two hypotheses might also act in a reciprocal way such that health affects socioeconomic status and socioeconomic status affects health. For example, based on data from a nationally representative longitudinal survey, Mulatu and Schooler find that both social causation and health selection hypothesis contribute to social inequalities in health, though the causal path from socioeconomic status to health status is stronger than the reverse [20].

Though scholarly interest has focused on explaining the topic according to those two hypotheses, few studies have explored the relationship between self-rated health and the subjective perception of social mobility. Social mobility is “a study of change, of movement” [21] (p. 1). It refers to the movement in social status relative to one’s current social location within a given society. Lipset and Bendix’s classical research finds that the downwardly mobile do not identify with their destination class and tend to be more conservative in terms of political attitudes: “In all countries, manual workers coming from middle-class backgrounds should be expected to desire a return to the higher class, and hence should be likely to retain middle-class values and patterns of behavior.” Similarly, the upwardly mobile “who rise to middle-class status become politically conservative” [22]. The underlying hypothesis is that people in general prefer to adopt the more prestigious identity and thereby to maximize their status, including their health status [23,24,25,26]. 

Such phenomena can be explained by an individual’s degree of life satisfaction, which derives primarily from a subjective sense of social class and mobility. For example, researchers have found that high-performing students usually rate their health higher than do those with low grades [27]. Similarly, people who receive salary raises and promotions tend to perceive their health conditions as good [28]. As noted in past research, most people have little or no sense of a social distribution system or the inequality or deprivation within it; satisfaction with life comes from comparing their own social rank and health with those of their local cohort group. As Norton states, “the broad shape of an overall distribution of outcomes matters much less than the local shape of an individual’s most salient distributions [29].” In this view, those with higher life satisfaction are those who see themselves as having a better chance for upward mobility [26,30,31]; however, whether such results are applicable in a global context is still unknown.

In this paper, we take a social causation approach to engage with theoretical debates and empirically examine how perception of class mobility will affect an individual’s self-rated health in China. Using eight waves of Chinese General Social Survey data spanning ten years between 2005 and 2015, we conducted the analysis at the micro (individual) level and at the macro (provincial) level. The results from both levels yielded consistent results. At the individual level, we found that those who perceived themselves as experiencing upward mobility are positively associated with a higher health status compared with those who perceived themselves as experiencing downward mobility. At the provincial level, our findings from static panel analysis further revealed the positive relationship between the perception of class mobility and provincial health level.

## 2. Data, Variables, and Methods

### 2.1. Data and Sample

We conducted this research using two types of analytical sample: the individual-level sample and the provincial-level sample. The primary data source was the Chinese General Social Survey (CGSS), a national representative survey using multistage stratified national probability sampling. Officially launched in 2003, CGSS aims to systematically monitor the changing relationship between social structure and quality of life in urban and rural China. Multilayered strata sampling is used in surveys and samples drawn from each stage, including primary sampling units from districts, secondary sampling units from neighborhood residential areas, and tertiary sampling units from residents’ committees. One eligible person, aged eighteen or over, was randomly selected from each sampled household to serve as the survey respondent. Each wave of the CGSS covered between about 5000 and 12,000 households from twenty-six to thirty-one provinces and collected comprehensive information such as individuals’ self-rated health, employment status, and various social attitudes [32]. The survey response rate was over 50%.

For the individual-level analysis, we pooled eight waves of CGSS spanning the years 2005 to 2015 (e.g., CGSS 2005, 2006, 2008, 2010, 2011, 2012, 2013, and 2015). CGSS adopted repeated cross-sectional design, each wave was thus administered to a new sample of interviewees at successive time points. In this vein, the CGSS samples involve largely different individual people for each survey year. Owing to the sample design, household sizes varied among respondents, though individuals from large families had a higher probability of being selected into the sample. Moreover, there were differences between the actual sample size and the real annual population in China from 2005 to 2015. To correct for these two major sources of biases, we first computed household weights for each sample to achieve a representative figure for the annual general population. We then computed the population weights for rural and urban samples and further normalized the population weight by using the real population size of each survey year so that all waves of data correctly reflected the real population in China. After excluding the missing values on some key independent variables—in particular, familial income and employment status—our working sample for multivariate regression shrank from 78,097 to 65,829. 

For the provincial-level analysis, we computed average health status, average score of self-perceived class mobility, and subjective social class for each province from the eight-wave CGSS data and merged that with corresponding provincial-level socioeconomic statistics from the China Statistical Yearbook, published by National Bureau of Statistics of China. The construction of data as such constituted province panels. This sample contains twenty-five provinces; for each province, there are eight observations. 

### 2.2. Variables

For individual-level analysis, self-rated health is the dependent variable. This was assessed by asking respondents, “In general, how would you rate your health?”, with the options being, 1 (“very unhealthy”), 2 (“unhealthy”), 3 (“so-so”), 4 (“healthy”), and 5 (“very healthy”). Though self-rated health may be a problematic measure since it primarily reflects an individual’s subjective evaluation of his or her own health, existing literature has widely shown that this measure has robust predictive power on mortality, morbidity, and individuals’ physical functioning [33]. The main explanatory variable—subjective class mobility—is captured by a single question in 2005 and 2006: “Compared with three years ago, how would you describe your current socioeconomic status?”, with three ordered responses—“lower,” “almost the same,” and “higher.” For the rest of years (i.e., 2008, 2010, 2011, 2012, 2013 and 2015), we constructed the subjective class mobility based on two questions, 1) which level do you think you should belong to? 2) which level do you think you should belong to 10 years ago? each question contains 10 scales, where 1 represents the highest and 10 represents the lowest. We calculate the subjective class mobility by taking the difference between these two questions. We rate subjective class mobility as lower, if Q1−Q2 is positive, higher if Q1−Q2 is negative, and almost the same if Q1−Q2 is zero (note that, though the measure of subjective class mobility shows some inconsistency, the reference years are all within the period 2000–2005, when the socio-economic development began to revive after large-scale laid off in China. To check the robustness of the results, we also conducted the analysis based on 2008–2015 seven-year data to rule out the measurement biases, the results are consistent (not shown, but available upon request). To reveal the effect of self-perceived class mobility on an individual’s self-rated health, we controlled for a series of individual and familial characteristics that affect self-rated health and that may also be associated with self-perceived class mobility. These control variables include age, years of schooling, familial annual income, a five-category subjective social class (1 = lower, 2 = lower middle, 3 = middle, 4 = upper middle, 5 = upper), communist party membership (1 = yes, 0 = otherwise), urban registration status, known as urban *hukou* (1 = yes, 0 = otherwise), gender (1 = male; 0 = female), marital status (1 = married, 2 = never married, 3 = separated/divorced/widowed), work status (0 = unemployed, 1 = employed, 2 = retired), and wave dummies. Descriptive statistics are shown in Table 1. 

For the provincial-level analysis, the dependent variable of interest was provincial average health status and the key independent variable was the average score of class mobility. Other control variables included the provincial average score of subjective social class, logged value of number of college students in the survey, logged gross domestic product (GDP) per capita, logged GDP annual growth rate, logged wage level, Gini index, logged number of non-governmental organization (NGO) associations, provincial proportion of party members, particulate matter ten micrometers or less in diameter (PM_10_) as a measure of air quality, and logged number of hospital beds. These variables, derived from the aspects of economic development, social development, political party structure, the environment, and healthcare availability, attempt to provide explanations for the spatial variation and temporal trend of health status. For the description of variable distribution, please refer to Table 2. 

### 2.3. Analytical Procedures

The overall analysis was conducted using Stata/SE, version 14.2 (StataCorp LP., College Station, TX, USA). For individual samples, since self-rated health is an ordinal variable, we adopted ordered logistic regression for this part of the analysis. Considering the large rural-urban divide in China, first conducted the analysis on full sample, and then run the models for rural and urban sample respectively. For the provincial sample, considering the provincial panel design, we conducted panel data analysis. Since the lagged dependent variable did not show significant effects—meaning that present health status was not affected by past health status—static panel models were employed for this part. In part, it is because CGSS employed repeated cross-sectional design, rather than a longitudinal design. For these, we adopted pooled ordinary least squares (OLS) regression, a random-effects model, and a fixed-effects model; time-fixed effects are controlled for all three models. When conducting pooled OLS, we assumed provincial homoskedasticity and largely ignored the province-specific effects, which is often the case in panel setting. To accommodate provincial heterogeneity, more advanced techniques, such as a random-effects model and a fixed-effects model, are further considered. The model can be written as follows:(1)$$Health_{it}=\beta Mobility_{it}+X_{it}\gamma +\alpha_{i}+u_{it};\;i=1,2,\ldots ,\;5;\;t=1,2,\ldots ,\;8.\;$$
where $Health_{it}$ is the dependent variable—provincial health level in province *i* at time *t*; $Mobility_{it}$ is the key independent variable—subjective class mobility score in province *i* at time *t*; β is the corresponding coefficient of $Mobility_{it}$; $X_{it}$ represents a variable matrix containing various other time-variant provincial characteristics, including provincial subjective social class, economic development, social development, political party structure, environmental, and medical supply measures (see Table 2); γ is the corresponding coefficient vector of $X_{it}$; $\alpha_{i}$ is the unknown province-specific intercept, which is time-invariant; $u_{it}$ is the time-variant error term. In pooled OLS, we mixed $\alpha_{i}$ and $u_{it}$ and assumed that the composite errors $v_{it}$ ($v_{it}=\alpha_{i}+u_{it}$) are homoscedastic and independent of each other. However, if $\alpha_{i}$ is correlated with one or more $X_{it}$—that is, $\mathrm{cov}\left(\alpha_{i},X_{it}\right)\neq 0$—OLS would produce a biased and inconsistent estimation. If $\alpha_{i}$ is uncorrelated with $X_{it}$—that is, $\mathrm{cov}\left(\alpha_{i},X_{it}\right)=0$—and meanwhile we have more than one observation on each province, OLS would produce unbiased yet inefficient estimates. When running the random effects model, we assumed that the expected value of errors would be zero, that is, $E\left(\alpha_{i}+u_{it}|X_{it}\right)=0$, or $\mathrm{cov}\left(\alpha_{i},X_{it}\right)=0$; if this assumption would have been violated, the random-effects model would have produced inconsistent estimates [34]. For the fixed-effects model, we allowed $\mathrm{cov}\left(\alpha_{i},X_{it}\right)\neq 0$, since the province-specific effect is time invariant and can be taken as the intercept. 

We conducted various statistical tests to choose the best-fitting model. Relative to pooled OLS, whether the fixed-effects model is preferred can be tested by an F test, and whether the random-effects model is favored can be tested by Breusch-Pagan Lagrange multiplier (LM) test. For the above two comparisons, insignificant results imply the preference for pooled OLS over fixed-effects or random-effects models. To choose between random-effects and fixed-effects models, we conducted a Hausman test; if there were significant differences in estimates, we preferred the fixed-effects model; otherwise, we chose the random-effects model. 

## 3. Results

### 3.1. Distribution of Health Status

Figure 1 depicts the provincial snapshot of health status in 2005 and 2015; the darker the color, the better the health. As shown, in 2005, the health status is quite scattered, while in 2015, there seems to have been declining health in eastern and western areas except Yunnan, Guizhou, and Guangxi, three provinces with more areas of higher elevation. In general, from 2005 to 2015, the average health status showed some improvement, particularly in the middle and lower reaches of the Yangtze River. 

To visualize how self-perceived class mobility is associated with individual health status, Figure 2 is drawn to show the distribution of health status across different types of mobility patterns. From left to right, each bar presents the perception of downward mobility, immobility, and upward mobility. For each bar, there are five segments; these segments add up to 100% for that specific mobility pattern. 

For example, it is shown that in China, among those who perceived themselves to be experiencing upward mobility, 21.59% rated themselves very healthy, 36.34% rated themselves healthy, and only 18.99% rated themselves unhealthy or very unhealthy. If further looking at rural and urban sample, there are 62.47% urban residents in this group rated themselves as healthy and very healthy, 13.74% more than those in rural areas. However, for those who perceived themselves to be downwardly mobile, there are 12.67% of people rated themselves as very healthy, 29.08% as healthy, and 22.73% as unhealthy or very unhealthy in China. When looking at rural and urban sample respectively, it is shown that among those who perceived themselves to be experiencing downward mobility, 27.51% of rural residents considered themselves to be unhealthy and very unhealthy, 35.3% more than urban areas.

### 3.2. Individual-Level Analysis

To examine the effect of self-perceived class mobility on an individual’s health, we first took health status as an ordinal variable and ran ordered logistic regression based on individual-level data. Table 3 presents the results of this ordered logistic regression, cutpoints shown at the lower panel of the table refer to the estimation of ordered categories of the health. For all three models, provincial and time dummies are controlled. Model 1 is the baseline model, in which we included only for class mobility. It is shown that the odds of being very healthy and healthy versus the combined so-so, unhealthy, and very unhealthy were about 19.12% (=$\;e^{0.175}-1$) higher for those who perceived themselves as having no mobility and 36.21% (=$\;e^{0.309}-1$) higher for those who perceived themselves as experiencing upward mobility, compared to downward mobility. 

When examining class mobility, there has been a long-standing debate about whether holding low-wage jobs improves the chances of upward mobility [35,36,37]. The mixed results in previous studies imply that subjective evaluation of class may be a potential factor to consider. In Model 2, we accounted for self-rated class, a factor not only highly correlated with self-perceived class mobility, but also self-rated health level. As shown, when including subjective class, the magnitude of the self-perceived mobility reduced to about half compared to Model 1. When further controlling for a series of individual and familial characteristics, such as residence, age, gender, party membership, marital status, work status, years of schooling, and total family income, that have proved to be central in predicting self-rated health in previous studies in Model 3, the effect of self-perceived mobility only slightly decreased. The results show that compared to those who consider themselves as experiencing downward mobility, the odds of being very healthy and healthy versus the combined so-so, unhealthy, and very unhealthy were about 6.29% (=$\;e^{0.061}-1$) higher for those who perceived themselves as having no mobility and 11.29% (=$\;e^{0.107}-1$) higher for those who perceived themselves as experiencing upward mobility, holding constant of other factors. Even when switching the reference category to immobility, those who perceived themselves as having upward mobility were still more likely to rate themselves as significantly healthier. Moreover, the results of residence suggest that, urban residents rated themselves healthier compared with rural residents. As rural and urban areas are often associated with different class structure and mobility chances, in order to show how the perceptions to class mobility vary in rural and urban samples, we further run the regression for rural and urban samples respectively. From Table 4, it is clear that self-perceived class mobility was more strongly associated with self-rated health in rural areas than urban areas. It is shown, the odds of being very healthy and healthy versus the combined so-so, unhealthy, and very unhealthy were about 13.5% (=$\;e^{0.127}-1$) higher for those who perceived themselves to upwardly mobile in rural areas, compared to those who consider themselves as experiencing downward mobility. However, the relative odds in urban areas was only 5.94% (=$\;e^{0.057}-1$), and it is only marginally significant. If taking immobility as the reference group, self-perception of experiencing upward mobility was significantly associated with higher health level in rural areas, but not in urban areas (not shown, but available upon request). 

Recent scholarship has proposed that it is often problematic to compare log-odds or odds ratios across models with different predictors in logistic regression, since unobserved heterogeneity tends to vary across models [38]. We thus present the average partial effect of self-perceived class mobility on the probability of rating oneself as very unhealthy, unhealthy, so-so, healthy, and very healthy in Table 5. This result is calculated based on Model 3 in Table 3 and by-residence results in Table 4. As shown, perceiving oneself as remaining stable compared to experiencing downward mobility would on average reduce an individual’s probability of reporting lower health—by 0.002 for “very unhealthy,” by 0.006 for “unhealthy,” and by 0.004 for “so-so”—and would increase the probability of an individual reporting better health— by 0.004 for “healthy” and by 0.008 for “very healthy.” Such partial changes are prevalent in rural areas, not urban areas. It is also the case when comparing those who perceived themselves as experiencing upward mobility versus downward mobility or immobility, the significant partial changes in probability of $Health_{i}$ (i = 1,2,3,4,5) were significant only in rural areas. In urban areas, such partial changes in probability of $Health_{i}$ (i = 1,2,3,4,5) either had marginal significance, or no significance. Figure 3 is further drawn to illustrate such pattern. 

### 3.3. Provincial-Level Analysis

To investigate whether the pattern persists at the macro level, we also conducted a provincial panel data analysis. We first fit a pooled OLS model (Model 1). Consistent with individual-level analysis, the results showed that at the provincial level, average self-perceived mobility was positively associated with health score, controlling for a series of other provincial characteristics. It was also found that political party structure (measured by proportion of communist party membership) and economic development (measured primarily by logged GDP per capita) both negatively predicted health status at the provincial level. While somewhat unexpected, the negative relations might reflect the fact that development does not necessarily bring about health benefits; for instance, fast food restaurants, which pose threats to public health, have shown strong growth in regions with higher levels of development in many developing countries. These regions often have a high prevalence of obesity, which is associated with increased risks of such concurrent morbidities of chronic diseases as type II diabetes, insulin resistance, coronary artery disease, hypertension, and osteoarthritis [39,40]. 

Although the OLS is based on the provincial homoskedasticity assumption, this is often not the case for panel data design. To accommodate for province-specific effects, we then employed a random-effects model by allowing intercepts to vary by province; the results are shown in the middle panel of Table 6. Clearly, the results of random-effects and OLS regression showed great similarity. To choose between these two models, we conducted a Breusch-Pagan LM test, whose results suggested that we cannot reject the null hypothesis that the variances across provinces was zero (*p* > 0.1), meaning that pooled OLS regression was preferred. However, we could not rule out the possibility that there may have been some omitted variables at the province level that could have affected the self-perceived class mobility level as well as the health level; for this reason, we employed a fixed-effects model, whose results are presented in the last column of Table 6. By using a fixed-effects model, we removed the time-invariant provincial characteristics—observed or unobserved—and assumed that within-province variation is vital in predicting the provincial health level. Despite some reduction of coefficient magnitude, the significance of the variables barely changed. Given that all the other variables were controlled, an increase in average self-perceived mobility score was associated with a 0.156 increase in provincial health level. Besides, across all three models, number of college students have shown consistent positive effect on health score. As a measure of educational development, such effect implies that higher educational level is conducive to develop effective abilities, shared lifestyles and cultural value attitudes which help avoid health risks, leading to such positive relationship [41,42]. Overall, the results of an F test suggested the null hypothesis that all within-province errors are equal to zero has been rejected (*p* < 0.001), meaning that a fixed-effects model is preferred in this analysis.

## 4. Discussion

This study contributes to the extant literature by exploring the link between the perception of class mobility and self-rated health. While numerous studies have shown the importance of an individual’s socioeconomic status on his or her self-rated health, few have investigated how self-perceived class mobility, a measure highly correlated with an individual’s de facto social class and previous mobility experiences, is associated with an individual’s self-rated health. To fill this gap, we employed eight waves of Chinese General Social Survey Data spanning the years 2005 to 2015 to conduct the analysis at the micro level (the individual) and the macro level (the province). Analyses at both levels yielded consistent results. At the individual level, we found that perception of experiencing downward mobility was associated with significantly lower self-rated health in both rural and urban areas when comparing with those who consider themselves to be upwardly mobile or immobile. However, perceived upward mobility may not necessarily improve individuals’ self-rate health level, especially in urban areas. At the provincial level, our findings further showed that the higher the score of perceived class mobility, the better the health status. 

For years, the ecological fallacy, a formal fallacy in the interpretation of individual-level behavior from aggregate-level data, has been the most persistent statistical problem in social science. A key point is that in the process of aggregation, micro-level information would inevitably be lost, making conclusions from micro-level (individual) data unreliable if directly recovered from the aggregate level. However, such inference can be achieved when the certain assumption is met. With respect to the present case, it is reasonable to believe that both levels can suggest a similar pattern if an individual’s perception or attitudes toward his or her own class mobility remain relatively independent, or has no spill-over effects across provinces. This assumption is proper, since mobility perception is too personal to be contagious and further alters the overall distribution of class mobility. In any case, our research focus is not to make any inference from the macro to the micro level or vice versa. Rather, we have attempted to examine the degree to which the macro-level conclusion matches with the micro-level before negating the pattern from any level. Though ad hoc, investigations as such can facilitate new knowledge-building and complement the extant studies by prying into the level of consistency at the micro, macro, or even meso levels. 

Still, this research has some limitations that deserve further discussion. First, though we identified a positive correlation between self-perceived class mobility and self-rated health at both the individual and provincial levels, we cannot identify through which specific channel this linkage is formed. Second, mobility in this analysis primarily refers to intragenerational mobility. By definition, intragenerational mobility involves secular changes of an individual’s economic position, or specifically, the occupational standing across time. However, based on the existing data and operationalization, distinguishing actual moves from temporary changes in socioeconomic status given the time period is like finding a needle in a haystack, making the genuine effect of mobility almost unrecoverable. Besides, our analysis assumes that there is a notable consistency between perceived mobility and de facto mobility; however, the potential disjunction between those who perceive themselves as moving but who remain stable, and vice versa, cannot be ignored. Third, scholars have shown that an individual’s chances for mobility are largely associated with that individual’s socio-economic status or similarly, class position. Though our analysis accounted for subjective class position which is highly associated with one’s de facto socio-economic status, mobility process is by no means random, which cannot be solved solely by controlling for cross-sectional subjective evaluation. In this vein, we have revealed only the association between these two. 

Despite those limitations, to the best of our knowledge, this is the first study that attempts to link subjective class mobility to health. We hope that this research can be used as a first step to a more robust future investigation that might uncover the specific mechanism of that linkage.

## 5. Conclusions

This analysis contributes to the extant literature in two ways. First, we have provided the evidence that the self-perceived class mobility, a measure correlated with one’s de facto social class but has been largely overlooked in existing studies, can predict one’s self-rated health. Second, such pattern has shown some consistency from both micro and macro levels. Though interpreting micro-level behavior from macro-level information may suffer from the ecological fallacy, it is important to examine in what circumstance, and at what degree can micro level match with macro level results, or vice versa. 

## Figures and Tables

**Figure 1 ijerph-15-02644-f001:**
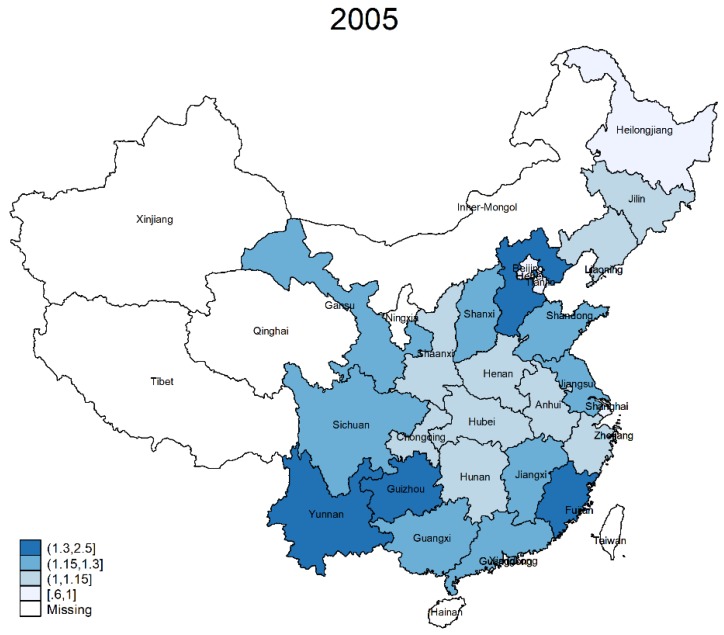
Geographic Distribution of Health in 2005 and 2015.

**Figure 2 ijerph-15-02644-f002:**
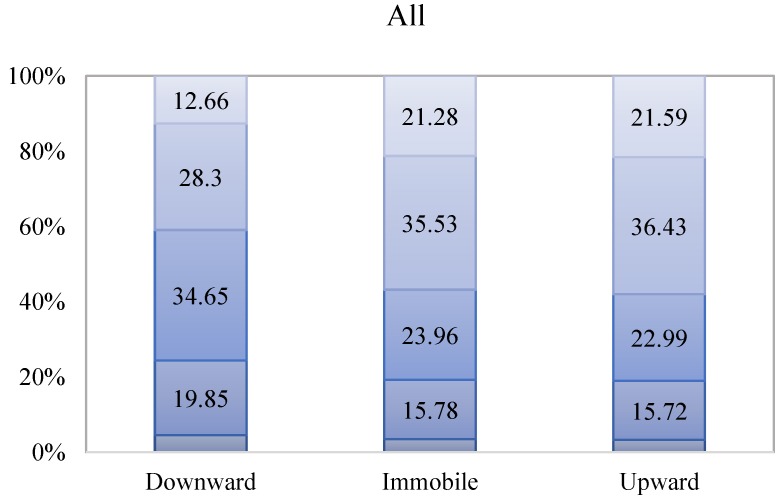
Distribution of Self-Rated Health by Self-Perceived Class Mobility.

**Figure 3 ijerph-15-02644-f003:**
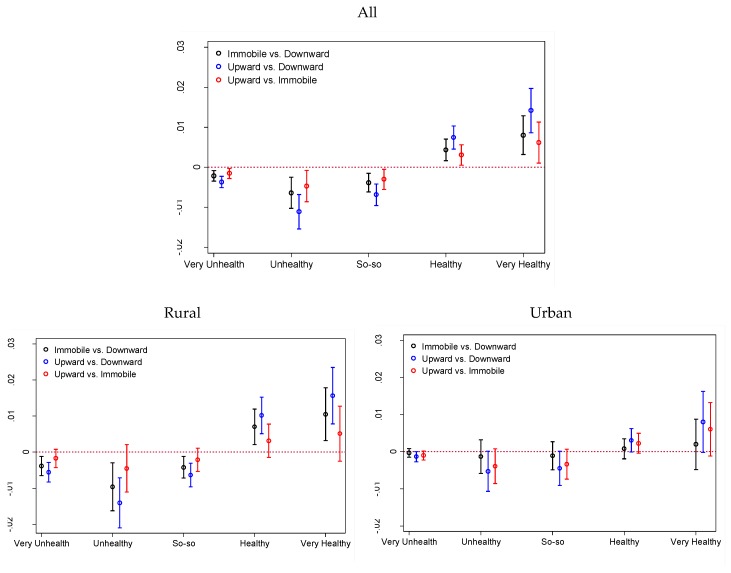
Average Partial Effect of Self-Perceived Class Mobility.

**Table 1 ijerph-15-02644-t001:** Descriptive Statistics for Selected Variables (2005–2015).

Variable		Proportion (%)	Mean (SD)
Dependent Variable			
Health	Very Unhealthy = 1	3.75	
	Unhealthy = 2	17.05	
	So-so = 3	26.94	
	Healthy = 4	33.53	
	Very Healthy = 5	18.72	
Independent Variable			
Mobility	Downward = 1	30.26	
	Immobile = 2	44.79	
	Upward =3	24.95	
Class	Lower = 1	20.37	
	Lower Middle = 2	31.43	
	Middle = 3	41.46	
	Upper Middle = 4	5.85	
	Upper = 5	0.90	
Party Member	No = 0	89.13	
	Yes = 1	10.87	
Hukou	Rural = 0	50.99	
	Urban = 1	49.01	
Gender	Female = 0	51.10	
	Male = 1	48.90	
Marital Status	Married =1	86.23	
	Never Married = 2	5.74	
	Divorced/Widowed = 3	8.03	
Work Status	Unemployed = 0	20.03	
	Employed = 1	61.92	
	Retired = 2	18.05	
Wave	Year 2005	12.31	
	Year 2006	12.17	
	Year 2008	12.67	
	Year 2010	11.77	
	Year 2011	11.99	
	Year 2012	11.83	
	Year 2013	13.62	
	Year 2015	13.65	
Age (years old)			45.93 (14.93)
Years of Schooling (years)			8.40 (4.30)
Family Annual Income (RMB)			41,608.78 (104,372.20)

Note: Data are weighted. Numbers in the parentheses are standard deviations. Percentages may not add up to exactly 100 per cent, owing to rounding off.

**Table 2 ijerph-15-02644-t002:** Descriptive Statistics for Selected Provincial-Level Variables (2005–2015).

Variable	Description	Mean	S.D.
Dependent Variable			
Health	Average Self-Rated Health	1.26	0.30
Independent Variable			
Class Mobility	Average Self-Perceived Mobility	0.72	1.71
Urbanization	Level of Urbanization	0.42	0.16
Class	Average Self-Perceived Class	1.47	0.35
Economic Development	GDP per capita (log value)	0.17	0.59
Economic Growth	GDP Annual Growth Rate (Log value)	0.13	0.06
Political Development	Percent of Party member	0.11	0.05
Social Development	Number of NGO Association (log value)	1.10	0.46
Inequality	Gini Coefficient (Household Income)	−0.78	0.18
Marketization	Marketization Index	1.85	0.25
Labor Cost	Average Wage Level (Log value)	10.30	0.80
Education	College Students per 100,000 People (Log value)	2.16	0.32
Environment	PM_10_	−2.31	0.26
Medical Care Development	Number of Hospital Beds (log value)	−5.61	0.28

**Table 3 ijerph-15-02644-t003:** Ordered Logit Model Predicting Self-Rated Health (2005–2015).

Variables	Model 1	Model 2	Model 3
Self-Rated Health			
Immobile	0.175 *** (0.019)	0.073 *** (0.019)	0.061 ** (0.019)
Upward	0.309 *** (0.020)	0.126 *** (0.021)	0.107 *** (0.021)
Urban Residence			0.123 *** (0.017)
Self-Rated Class (ref: Middle Class)			
Lower		−0.755 *** (0.021)	−0.561 *** (0.021)
Lower Middle		−0.366 *** (0.017)	−0.284 *** (0.017)
Upper Middle		0.163 *** (0.031)	0.118 *** (0.032)
Upper		0.235 ** (0.080)	0.447 *** (0.081)
Age			−0.041 *** (0.001)
Male			0.295 *** (0.015)
Party Member			0.059 * (0.025)
Marital Status (ref: Married)			
Never Married			0.083 * (0.033)
Separated/Divorced/Widowed			−0.043 (0.030)
Work Status (ref: Employed)			
Unemployed			−0.307 *** (0.021)
Retired			−0.099 *** (0.021)
Years of Schooling			0.018 *** (0.002)
Logged Family Income			0.139 *** (0.007)
Cut Point 1	−3.630 *** (0.048)	−4.025 *** (0.050)	−4.278 *** (0.093)
Cut Point 2	−1.645 *** (0.044)	−2.017 *** (0.046)	−2.146 *** (0.092)
Cut Point 3	−0.275 *** (0.044)	−0.622 *** (0.045)	−0.615 *** (0.091)
Cut Point 4	1.417 *** (0.044)	1.094 *** (0.046)	1.275 *** (0.091)
N	65,829	65,829	65,829
Log-Likelihood	−91,928.407	−91,118.680	−86,318.460

Note: i. adjusted robust standard errors in parentheses; ii. ** *p* < 0.01, *** *p* < 0.001 (two-tailed tests); iii. Province and wave dummies are controlled.

**Table 4 ijerph-15-02644-t004:** Ordered Logit Model Predicting Self-Rated Health by Residence (2005–2015).

Variables	Rural	Urban	Sig
Self-rated Health			
Immobile	0.086 ** (0.030)	0.014 (0.025)	*
Upward	0.127 *** (0.032)	0.057 ^†^ (0.030)	†
Self-Rated Class (ref: Middle Class)			
Lower	−0.513 *** (0.033)	−0.630 *** (0.029)	
Lower Middle	−0.272 *** (0.028)	−0.304 *** (0.022)	
Upper Middle	0.192 *** (0.055)	0.067 ^†^ (0.038)	*
Upper	0.697 *** (0.124)	0.089 (0.109)	***
Age	−0.039 *** (0.001)	−0.042 *** (0.001)	
male	0.345 *** (0.024)	0.214 *** (0.020)	***
Party Member	0.142 ** (0.048)	0.071 * (0.028)	
Marital Status (ref: Married)			
Never Married	0.080 (0.053)	0.057 (0.042)	†
Separated/Divorced/Widowed	−0.080 (0.052)	0.013 (0.036)	
Work Status (ref: Employed)			
Unemployed	−0.394 *** (0.037)	−0.316 *** (0.026)	*
Retired	−0.161 *** (0.041)	−0.158 *** (0.028)	
Years of Schooling	0.030 *** (0.004)	0.004 (0.003)	***
Logged Familial Income	0.168 *** (0.011)	0.090 *** (0.010)	***
Cut Point 1	−3.003 *** (0.177)	−5.572 *** (0.129)	
Cut Point 2	−0.927 *** (0.175)	−3.299 *** (0.126)	
Cut Point 3	0.516 ** (0.175)	−1.634 *** (0.125)	
Cut Point 4	2.341 *** (0.176)	0.334 ** (0.124)	
N	26,221	39,608	
Log-Likelihood	−35,257.311	−50,122.127	

Note: i. adjusted robust standard errors in parentheses; ii. ^†^
*p* < 0.1, * *p* < 0.05, ** *p* < 0.01, *** *p* < 0.001 (two-tailed tests); iii. Province and wave dummies are controlled.

**Table 5 ijerph-15-02644-t005:** Average Partial Effect of Self-Perceived Mobility.

Health	Immobile vs. Downward	Upward vs. Downward	Upward vs. Immobile
All			
Very Unhealthy	−0.002 ** (0.001)	−0.004 *** (0.001)	−0.002 * (0.001)
Unhealthy	−0.006 ** (0.002)	−0.011 *** (0.002)	−0.005 * (0.002)
So-so	−0.004 ** (0.001)	−0.006 *** (0.001)	−0.003 * (0.001)
Healthy	0.004 ** (0.001)	0.007 *** (0.001)	0.003 * (0.001)
Very Healthy	0.008 ** (0.002)	0.014 *** (0.003)	0.006 * (0.003)
Rural			
Very Unhealthy	−0.004 ** (0.001)	−0.006 *** (0.001)	−0.002 (0.001)
Unhealthy	−0.010 ** (0.003)	−0.014 *** (0.004)	−0.004 (0.003)
So-so	−0.004 ** (0.002)	−0.009 *** (0.002)	−0.002 (0.002)
Healthy	0.007 ** (0.003)	0.010 *** (0.002)	0.003 (0.002)
Very Healthy	0.011 ** (0.003)	0.016 *** (0.004)	0.005 (0.004)
Urban			
Very Unhealthy	−0.000 (0.001)	−0.001 ^†^ (0.001)	−0.001 (0.001)
Unhealthy	−0.001 (0.002)	−0.005 ^†^ (0.003)	−0.004 (0.002)
So-so	−0.001 (0.002)	−0.004 ^†^ (0.002)	−0.003 (0.002)
Healthy	0.001 (0.001)	0.003 ^†^ (0.002)	0.002 (0.001)
Very Healthy	0.002 (0.003)	0.008 ^†^ (0.004)	0.006 (0.004)

Note: i. adjusted robust standard errors in parentheses; ii. ^†^
*p* < 0.1**,** * *p* < 0.05, ** *p* < 0.01, *** *p* < 0.001 (two-tailed tests); iii. Province and wave dummies are controlled.

**Table 6 ijerph-15-02644-t006:** Static Panel Regression Predicting Provincial Health Level (2005–2015).

Variables	OLS	RE	FE
Average Self-perceived Mobility	0.167 *** (0.023)	0.167 *** (0.023)	0.156 *** (0.022)
Urbanization	−0.030 (0.075)	−0.030 (0.075)	−0.078 (0.071)
Average Subjective Social Class	0.364 *** (0.060)	0.364 *** (0.060)	0.218 *** (0.061)
Political Party Structure	−0.327 * (0.162)	−0.327 * (0.162)	−0.503 ** (0.164)
GDP per capita (log value)	−0.064 * (0.029)	−0.064 * (0.029)	0.053 (0.085)
GDP Annual Growth Rate (log value)	0.035 (0.164)	0.035 (0.164)	−0.256 (0.176)
Gini Index	−0.002 (0.030)	−0.002 (0.030)	0.001 (0.030)
Marketization	0.075 ^†^ (0.044)	0.075 ^†^ (0.044)	0.015 (0.081)
Average Wage Level (log value)	−0.026 (0.021)	−0.026 (0.021)	0.032 (0.021)
Number of College Students (log value)	0.313 *** (0.057)	0.313 *** (0.057)	0.337 *** (0.061)
PM_10_	0.023 (0.022)	0.023 (0.022)	0.020 (0.038)
Number of Hospital Bed (log value)	−0.033 (0.042)	−0.033 (0.042)	−0.092 (0.071)
NGO Association (log value)	0.028 (0.022)	0.028 (0.022)	0.017 (0.041)
Constant	−0.300 (0.241)	−0.300 (0.241)	−0.783 ^†^ (0.402)
N	200	200	200

Note: i. adjusted robust standard errors in parentheses; ii. ^†^
*p* < 0.1, * *p* < 0.05, ** *p* < 0.01, *** *p* < 0.001 (two-tailed tests). OLS denotes ordinary least square model, RE denotes random-effects model and FE denotes fixed-effects model.

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
