# Peer review of "The Health Consequences of Social Mobility in Contemporary China"

_ijerph, 2018, doi:10.3390/ijerph15122644_

Round 1
Reviewer 1 Report
This research uses data from the China General Social Survey (CGSS) to address a relevant and, to my knowledge, novel question regarding how perceptions of social mobility affect self-reported health in China. The dataset is comprehensive with representative survey data available across time and regions. The dual analyses of individual and aggregate data are interesting, as is the consistency of the findings across both analyses. However, the paper would benefit from some revisions to the introduction to better reflect the main focus of the study, as well as clarification of the class mobility survey item. The paper is potentially publishable subject to the questions and comments set out below: 1. Introduction: • Much of the introduction focuses on the social causation hypothesis versus the health selection hypothesis in explaining the well-established association between socioeconomic status and health. Although this is important background, the analyses reported focus on individual- and province-level data in relation to the research question, rather than assessing support for the social causation hypothesis versus the health selection hypothesis – as might be anticipated from the way the introduction is currently framed. • Is it also plausible that the two hypotheses act in a reciprocal way such that health affects SES and SES affects health? (e.g., Mulatu & Schooler, 2002) • Following on from the first two points, it appears that this study takes a social causation approach to the research question – a statement to this effect would help situate the study in the theoretical context described. • A brief paragraph on approaches to dealing with multilevel data to preface the use of OLS, random effects and fixed effects models in the analyses may be appropriate here? • In paragraph 5 of the introduction, beginning ‘for example’I would suggest moving the second sentence referring to life satisfaction to the start of this paragraph to improve the flow. • Including a conceptual definition of social mobility and stating the overall direction of findings for SES and health could also strengthen the introduction. 2. Self-perceived class mobility item: This is most likely a translation issue but checking the wording of the survey item on which self-perceived mobility is based would be helpful to clarify that the focus is on people’s perceptions of the direction of their mobility in the past three years rather than their feelings about their socioeconomic status. This is not clear as currently written: Compared with the past three years, how would you feel about your socioeconomic status? For example, and depending on the original survey wording of course, this may be something like “Compared with three years ago, how would you describe your current socioeconomic status?” Additional points: 1. Individual-level vs. province-level findings: In light of the ecological fallacy issue raised in the Discussion, would it be useful to highlight/expand the brief mention that province was controlled for in the individual analyses (3.2; Note 3 in Tables 3 and 4)? 2. Suggest presenting Model 4 (Table 3) in a separate table for clarity, if appropriate. 3. It may be helpful to explicitly state that the CGSS samples are different for each survey year and do not overlap. 4. An explanation of the finding in 2.3 that there were no significant effects for the lagged DV (i.e., that past health ratings were not associated with present health ratings) would be useful, in terms of whether this was expected and possible reasons. 5. Consider moving the rationale for inclusion of the covariates provided in 2.3 up to 2.2 where the covariates are introduced and provide references. 6. Figures 1 and 2 are informative. I wonder though, if Figure 1 would have more impact without the lines (the key should suffice); and for Figure 2 (and 3.1, para 2 in the text), it would be interesting to know the results of any significance testing for the group differences shown. Reference: Mulatu, M.S. & Schooler, C. (2002) Causal connections between socio-economic status and health: Reciprocating effects and mediating mechanisms. Journal of Health and Social Behavior, 43(1) 22-41.
Author Response
Thanks for your very helpful comments. We have substantially revised our manuscript based on your comments and suggestions. Attached please find a summary of the revisions we have made and our response to your comments.

Reviewer 2 Report
Many thanks for giving me the opportunity to review this manuscript examining the link between the perception of class mobility and self-rated health in China. The manuscript is timely and potentially interesting, and it suits to the Journal. More or less the paper has a standard format of the articles in epidemiological studies. Overall, I think the paper makes a contribution to the literature in the selected field.
Comment: The authors showed great attention to the methodological/statistical aspects (that’s good!) but not so much to the interpretation of findings.
As the text of the Results section is very condensed I have just few questions on the present version of the paper:
In figure 1, what is the meaning of lines joined in the upper left corner?
In table 3, what is the meaning of Cut Points 1, 2, … under “Occupational Dummies”? What values are presented in the table? Interpretation (lines 215 - 222) of results seems like interpretation of results of binary logistic regression analysis but not as interpretation of results derived from ordered logistic regression analysis.
In table 5, how you can explain the role of “Number of College Students” on the predicting provincial health level? (Correct the number of table in the text).
What is meaning of the index "i" in the equation (lines 162-163)?
Author Response

(The authors gave the same response as above.)

Reviewer 3 Report
The manuscript is well-written, the analyses are well done and the topic is highly relevant to the journal's audience.
Author Response
Thanks for your comments.